# The Role of Nitric Oxide in Cancer: Master Regulator or NOt?

**DOI:** 10.3390/ijms21249393

**Published:** 2020-12-10

**Authors:** Faizan H. Khan, Eoin Dervan, Dibyangana D. Bhattacharyya, Jake D. McAuliffe, Katrina M. Miranda, Sharon A. Glynn

**Affiliations:** 1Discipline of Pathology, Lambe Institute for Translational Research, School of Medicine, National University of Ireland Galway (NUIG), H91 YR71 Galway, Ireland; faizan.khan@nuigalway.ie (F.H.K.); e.dervan1@nuigalway.ie (E.D.); d.bhattacharyya1@nuigalway.ie (D.D.B.); j.mcauliffe1@nuigalway.ie (J.D.M.); 2Department of Chemistry and Biochemistry, University of Arizona, Tucson, AZ 85721, USA; kmiranda@email.arizona.edu

**Keywords:** nitric oxide, nitric oxide synthase, tumourigenesis, DNA damage repair, angiogenesis, epithelial to mesenchymal transition, immunomodulation, apoptosis, therapeutic target, cell cycle

## Abstract

Nitric oxide (NO) is a key player in both the development and suppression of tumourigenesis depending on the source and concentration of NO. In this review, we discuss the mechanisms by which NO induces DNA damage, influences the DNA damage repair response, and subsequently modulates cell cycle arrest. In some circumstances, NO induces cell cycle arrest and apoptosis protecting against tumourigenesis. NO in other scenarios can cause a delay in cell cycle progression, allowing for aberrant DNA repair that promotes the accumulation of mutations and tumour heterogeneity. Within the tumour microenvironment, low to moderate levels of NO derived from tumour and endothelial cells can activate angiogenesis and epithelial-to-mesenchymal transition, promoting an aggressive phenotype. In contrast, high levels of NO derived from inducible nitric oxide synthase (iNOS) expressing M1 and Th1 polarised macrophages and lymphocytes may exert an anti-tumour effect protecting against cancer. It is important to note that the existing evidence on immunomodulation is mainly based on murine iNOS studies which produce higher fluxes of NO than human iNOS. Finally, we discuss different strategies to target NO related pathways therapeutically. Collectively, we present a picture of NO as a master regulator of cancer development and progression.

## 1. Introduction

Nitric oxide (NO) is a biologically unstable, lipophilic, extremely diffusible, free radical that regulates numerous biological functions [1]. Since discovery of NO in the cardiovascular system as the endothelium-derived relaxing factor (EDRF), researchers have identified roles for NO in a wide range of pathophysiological processes [2]. Although dietary nitrate and nitrite (NO_2_^−^/NO_3_^−^) are precursors for a certain amount of NO biosynthesis, NO is generated primarily by three different NO synthase (NOS) enzymes: the neuronal (nNOS/NOS1), inducible (iNOS/NOS2), and endothelial (eNOS/NOS3) isoforms. In the presence of nicotinamide adenine dinucleotide phosphate (NADPH) and oxygen (O_2_), NOS oxidises L-arginine to form L-citrulline and NO (Figure 1a) [3,4]. Constitutively expressed eNOS and nNOS facilitate low levels of NO production in a calcium-dependent manner, while iNOS produces high levels of NO in a calcium-independent manner [3,5].

NO typically reacts facilely with other molecules that contain unpaired electrons such as free radicals and transition metals. In addition, the oxidation products of NO are also capable of interacting with other biological molecules and can result in toxicity [6]. NO takes part in biological reactions by three main processes:(a)Diffusion

As a small, neutral molecule, NO rapidly enters cells by simple diffusion. The main intracellular targets are heme proteins, but NO can also react with non-heme iron or protein-based free radicals such as the tyrosyl radical of ribonucleotide reductase, a rate limiting enzyme in DNA synthesis [7,8].
(b)Autoxidation

The reaction of NO with nitrogen dioxide (NO_2_) produce nitrous anhydride (N_2_O_3_), which is converted to nitrite in aqueous systems (Equation (1)) [9].
NO + NO_2_ ⇌ N_2_O_3_ + H_2_O → 2NO_2_^−^ + 2H^+^(1)
(c)Reaction with superoxide to form peroxynitrite

NO reacts with superoxide (O_2_^−^) to yield peroxynitrite (ONOO^−^) (Equation (2)). This species has varied reactivity, but a major biological pathway involves rapid interaction with carbon dioxide to form nitrosoperoxycarbonate (ONOOCO_2_^-^). While this reactive species leads mostly to nitrate (Equation (3)) [10], it can also generate the carbonate and NO_2_ radicals (Equation (4)), which can react with a wide range of biomolecules including nucleic acids, amino acids, lipids, metal containing proteins [11]. Protonation of peroxynitrite can also lead to radical formation through homolytic cleavage (Equation (5)), although, again, isomerisation to nitrate is the major pathway [12].
NO + O_2_^−^→ ONOO^−^(2)
ONOO^−^ + CO_2_→ [ONO_2_**.**CO_2_] → NO_3_^−^ + CO_2_(3)
ONOO^−^ + CO_2_→ [ONOOCO_2_^−^] → NO_2_ + CO_3_^−^(4)
ONOO^−^ + H^+^→ ONOOH → NO_2_ + OH(5)

The concentration and time-dependent effects of NO and NO-derived cellular adducts determine its biochemical and phenotypic consequences in cancer pathogenesis (Figure 1c) [3,13]. Low levels of NO (picomolar to nanomolar) have direct effects on cellular function by promoting cell proliferation and anti-apoptotic responses. The reaction between NO and organic free radicals or with metal complexes represents its direct effect [14]. High levels of NO (micromolar) have been shown to exert indirect effects and induce cell cycle arrest, apoptosis and senescence through both oxidative and nitrosative stresses [15]. The presence of other free radicals also influences NO signalling through reduction of the cellular NO level. Thus, the imbalance between the production and consumption rates of free radicals including reactive oxygen species (ROS) and reactive nitrogen species (RNS) can cause oxidative and nitrosative stress [16]. Cellular accumulation of ROS and RNS is involved in carcinogenesis. However, increased chemical stress may cause more deleterious effects in cancer cells such as protein modification, lipid peroxidation and DNA damage [17,18].

Although NO is a well-known signalling molecule, understanding the precise molecular effects it exerts will help elucidate its tremendous clinical potential in the diagnosis and treatment of cancer. Thus, here, we review the various actions of NO and its by-products in cancer and discuss how they may be leveraged as a novel therapeutic strategy in personalised cancer treatment.

## 2. NO Cell Signalling

NO-mediated pathways are regulated in either a cyclic guanosine monophosphate (cGMP)-dependent or cGMP-independent manner (Figure 1b) [3,19]. Although cGMP-dependent pathways have long been found to mediate several cellular processes, cGMP-independent reactions have also gained significant attention. These pathways involve different targets and varied modification processes.

NO produced by nNOS and eNOS typically exerts its effects via cGMP-dependent pathways [20,21,22], while the higher levels of NO produced by iNOS lead to protein modifications by RNS.

### 2.1. cGMP-Dependent Pathway

In the NO-mediated cGMP-dependent pathway, guanylate cyclase (GC) converts guanosine triphosphate (GTP) into the secondary messenger cGMP, which triggers the activation of cGMP-dependent protein kinases (PKG/PKA) [23], cGMP regulated phosphodiesterases (PDE) [24,25], and cyclic-nucleotide gated (CNG) ion channels [26]. These, in turn, impact vital biological processes, such as smooth muscle relaxation and regulation of blood pressure [27], platelet aggregation and disaggregation [28], and neurotransmission both peripherally [27] and centrally [29], in the processes of long-term potentiation and depression [29]. The involvement of cGMP in growth inhibition has been identified in vascular smooth muscle cells (VSMCs), in which GC interacts with NO produced by eNOS with a subsequent increase in cGMP and repression of the epidermal growth factor signalling pathway [30,31]. The increased cGMP phosphorylates target proteins such as the inositol 1,4,5-triphosphate receptor, resulting in a decrease in Ca^2+^ concentration and, in due course, smooth muscle relaxation [32]. The phosphorylation and subsequent degradation of β-catenin by protein kinase G (PKG) is also mediated by cGMP leading to the downregulation of growth-promoting and apoptosis-inhibiting proteins, including cyclin D1, c-myc, and survivin [33]. Similarly, NO produced by nNOS in neuronal cells of the retina and in the olfactory region attenuates the synaptic behaviour of NO-sensitive neurons where they are engaged in visual phototransduction and olfaction [34,35].

### 2.2. cGMP-Independent Pathway

The cGMP-independent pathway wields it effects through post-translational modifications of proteins [36,37], particularly at reactive thiol (RSH) and amine (RR’NH) sites, producing *S*-nitrosothiols (RSNO), and *N*-nitrosoamines (RR’NNO), respectively [38]. The *S*-nitrosation of cysteine residues has been found to regulate proteins and enzymes including NF-κB, AP-1 and CREB [39,40,41,42]. Various signalling cascades including G-proteins, the Ras pathway, mitogen-activated protein kinases (MAPK) signalling, and the phosphatidylinositol-3 kinase (PI3K) pathway are also regulated by *S*-nitrosation of their components [43,44,45]. *S*-nitrosation also regulates many NO-dependent signalling pathways, including Ca^2+^-dependent potassium channel induction in vascular smooth muscle [46], repression of mitochondrial cytochrome oxidase [47,48], bile acid secretion by Na^+^-taurocholate co-transporting polypeptide (NTCP) in hepatocytes [49], and calcium-release channels such as the ryanodine receptor [50]. RR’NNO can act as carcinogens. In particular, tobacco associated RR’NNO such as 4-(methylnitrosamino)-1-(3-pyridyl)-1-butanone (NNK) are key players in the development of lung cancer [51] and also promote the development of metastasis [52].

## 3. Role of NO in Cancer Biology

The role of NO in biological process depends on its source of generation, duration, and spatial concentration. At lower concentration, NO exerts cytoprotective effects and triggers carcinogenesis through activation of oncogenic pathways. However, at higher concentration, NO has been shown to produce cytotoxic effects in cancer cells and induce apoptosis [53]. The pro- and anti-tumour effects of NO in cancer biology play important roles by shifting the cellular response to stressors such as DNA damage, oncogene activation, altered cell metabolism and deregulating DNA repair enzymes and tumour suppressor genes, in addition to modulation of apoptotic and metastatic processes [54].

### 3.1. Genotoxicity and Mutagenesis

NO-mediated genotoxicity occurs due to the deamination of DNA bases, oxidation of bases and deoxyribose, strand breaks, and multiple types cross-linking events [55]. These reactions can result in direct genotoxicity induced by the deamination of amines with N_2_O_3_ and formation of peroxynitrite [56], or by indirect genotoxicity due to nitrosamine activation [57], apoptosis [58], DNA repair enzyme inhibition [59], or lipid peroxidation-induced DNA damage [60].

NO oxidises to form RNS that nitrosate primary heterocyclic amines of DNA bases (Equation (6)). Such nitrosated amines are susceptible to rapid deamination through isomerisation to a diazohydroxide followed by dehydration to a highly reactive diazonium ion (Equation (7)) [61,62]. Deamination plays a key function in the deleterious consequences of NO that could be important to the mutational landscape and heterogeneity observed within tumours. Deamination of guanine to xanthine causes depurination to create abasic sites in DNA, resulting in single strand breaks or misrepair [62]. NO-induced mutations were shown to be induced in TK6 cells, calf thymus DNA, yeast RNA, and bovine liver transfer RNA following NO treatment in vitro, resulting in the formation of a significant amount of xanthine and hypoxanthine [62]. Furthermore, macrophages stimulated with LPS and IFN-γ significantly induced the production of xanthine that could be repressed by the NOS inhibitor *N*-methyl-l-arginine monoacetate (NMA) [63].
RNH_2_ + N_2_O_3_→ RHNNO + NO_2_^−^ + H_+_(6)
RHNNO → RNNOH → RN_2_^+^ + H_2_O → ROH + N_2_ + H^+^(7)

One mutagenic example of NO in mammalian cells its targeting codon 248 (CGG) in the *p53* gene, which is one of the most commonly mutated genes in cancer [64]. Methylation of cytosine followed by its deamination results the formation of thymine [65]. The mutagenicity of NO has been demonstrated to cause deamination in CpG sites of the *p53* gene, resulting in a G-C → A-T transition, which plays a crucial role in different human cancers including colon, liver, breast, and lung [61,66]. The pattern of NO-induced point mutations was measured in the *supF* gene via deamination of either A or G that induce A:T → G:C transitions followed by G:C → A:T transitions [67].

The reaction of N_2_O_3_ with secondary amines can form carcinogenic *N*-nitrosamines that metabolise to strong alkylating electrophiles that can alter the N-7 and O-6 positions of guanine and the N-3 position of adenine in DNA. Alteration at the O6 position in guanine is prone to mutation and causes G→A transitions during DNA replication [68].

NO_2_ is also an NO-derived strong oxidant, which oxidises protein and non-protein thiols [60], protein sulphides [69], lipids [70], and low density lipoproteins [71]. Exposure of DNA to NO_2_ leads to the formation of 8-hydroxydeoxyguanosine [72] and 8-nitroguanine from guanosine [73]. Reaction of NO with O_2_^-^ can lead to medication of phenolic compounds to form highly mutagenic nitrated and hydroxylated product such as 8-nitroguanine and 4,5-dihydro-5-hydroxy-4-(nitrosooxy) guanine, which depurinate rapidly and cause transversion mutations (mainly G→T) at G-C pairs [74,75]. However, trace metals can change the chemistry to produce nitration and/or hydroxylation of DNA bases [75]. NO-induced oxidative deamination in the DNA of activated macrophages has been quantified by measuring 8-oxoguanine and 5-(hydroxymethyl) uracil formation [76]. The role of NO in both deamination and oxidation reactions was confirmed by NOS-inhibition, which suppressed both xanthine and 8-oxoguanine formation [63]. Intracellular NO_2_ formation can also induce DNA single-strand breaks either by direct DNA nicking [77] or involve other cellular factors such as abasic repair enzymes (i.e., activation of exonucleases and/or suppression of ligase and/or polymerases) [78,79].

The genotoxic effects of NO were suspected due to the fact that fluxes of O_2_^-^ form much more quickly than NO fluxes in activated macrophages and neutrophils. The genotoxic potential of NO metabolites has been confirmed by NOS inhibitors that prevented DNA damage in activated macrophages. The reaction of RNS with sulphhyldryl-containing peptides forms nitrosothiols, altering the function of ion channels [80], p21ras [81], protein tyrosine phosphatates [82], and cyclooxygenases [83]. S-Nitrosation of glutathione (GSH) induces apoptosis in epithelial cells [58], macrophage-like cells such as RAW 264.7 [84] and renal mesangial cells [85] by induction of nuclear chromatin condensation, DNA fragmentation and p53 accumulation. The formation of the malondialdehyde (MDA), a lipid peroxidation product induced by peroxynitrite is mutagenic in bacterial and mammalian systems and causes frameshifts and base-pair substitutions [86]. DNA damaging activity of NO and its metabolites was also increased due to inhibition of DNA repair proteins such as zinc finger-containing formamidopyrimidine DNA glycosylase (Fpg protein) [59] and O6-methylguanine -DNA-methyltransferase [87] by *S*-nitrosation of proteins.

### 3.2. DNA Damage Repair (DDR)

As described above, NO and its derivatives induce DNA damage leading to activation of DDR signalling networks [88]. The DDR response is regulated by five main DNA repair mechanisms that include nucleotide excision repair (NER), base excision repair (BER), mismatch repair (MMR), and the two double strand break repair pathways: non-homologous end-joining (NHEJ) and homology directed repair (HDR) (Figure 2) [89,90]. The NHEJ and HDR are also involved in the Fanconi Anaemia (FA) pathway to achieve interstrand DNA crosslink repair [91]. More than 150 proteins have been identified in DDR mechanisms, which control the progression of neurological diseases [92], aging [93], cancer risk [94], cancer therapy outcomes [95], inflammation, and other genetic syndromes with a variety of distinct phenotypes [96].

The NER pathway is a complex process involving more than 30 proteins that work to eliminate bulky DNA lesions by special endonucleases (Figure 2b) [97]. NER includes two distinct pathways, i.e., global genome-NER (GG-NER) and transcription coupled-NER (TC-NER). GG-NER is regulated by the xeroderma pigmentosum group c protein, which recognises DNA damage and recruits other repair machinery, while TC-NER is initiated by blocking RNA polymerase II activity at altered areas undergoing transcription. Both GG-NER and TC-NER involve the basic mechanism: (1) recognition of the damaged sites (in transcribed and non-transcribed regions of the genome, respectively); (2) verification of DNA damage; (3) excision of damaged oligomers; and (4) gap filling by ligating intact molecules [90]. Defects in the NER pathway causes rare autosomal recessive diseases such as Xeroderma pigmentosum (XP), Cockayne syndrome (CS), and trichothiodystrophy (TTD) [98,99,100]. NO was shown to inhibit DNA-adduct NER in human fibroblasts, including UVC and cisplatin derived DNA adducts [101]. Treatment with iNOS inhibitors increased NER activity. In contrast, NO did not impact BER in the same system.

The BER pathway specifically repairs *N*-alkylation DNA damage such as N7MeG, N3MeA, and N3MeG, in which DNA glycosylases search the damage sites in the genome, creating an apurinic/apyrimidinic (AP) site. Further, AP-endonuclease (APE) cleave the DNA at AP sites to form one nucleotide gap which is filled with the correct nucleotide by polymerase-β. NO has previously been reported to induce nuclear export of APE to the cytoplasm, via *S*-nitrosation of APE1 at cysteine 93 and 310, which may disrupts its DNA repair function [102]. Finally, DNA ligase IIIα complex containing XRCC1 protein seals the left-over nick in the DNA backbone (Figure 2a) [103]. Defects in this BER enzyme are associated with premature aging, cancer, and neurodegenerative diseases [104,105]. However, the exact mechanisms of BER dysregulation are not well known. Thus, downregulation of BER components has gained significant clinical interest in cancer therapy, using small molecular inhibitors in combination with radio- and/or chemotherapeutic agents [106,107]. iNOS overexpression in cholangiocarcinoma cells was found to inhibit 8-oxodeoxyguanine base excision DNA repair [108]. The inhibitory effect of iNOS could be reversed with NO scavengers but not sGC inhibitors indicating that this was not mediated by cGMP-dependent NO signalling [108].

Nitrosative deamination of guanine gives rise to two products, xanthine (Xan) and Oxa. Oxa can further react with spermine to form Oxa-spermine cross-link adducts (Oxa-Sp). Nakano et al. examined the ability of NER and BER enzymes to repair Oxa and Oxa-Sp in *E. coli* and human systems and found that Oxa and OXA-Sp DNA were preferentially repaired by NER, with BER enzymes only having weak reparatory activity. This suggests that the nature of NO DNA adducts may influence which excision repair pathway works optimally [109]. Mutumba et al. (2011) found that XRCC1 facilitates alkyl adenine DNA glycosylase (AAG) initiated excision of two key NO-induced DNA lesions: 1,N(6)-ethenoadenine and hypoxanthine, indicating that depending on the type of DNA damage that occurs BER can be activated in response to NO related DNA damage, while its effectiveness also depends on the type of damage involved (guanine versus adenine) [110]. *S*-nitrosation of AAG has also been found to increase the activity of AAG [111]. This coupled with the impact of NO on APE export to the cytoplasm may lead to BER enzymatic machinery imbalance, disrupting the effectiveness of BER in NO related DDR.MMR fixes replication associated errors that can arise due to insertion, deletion, and mis-incorporation of bases in the newly synthesised strand during DNA replication (Figure 2c) [112]. The MutSα complex (MSH2 and MSH6) or MutSβ complex (MSH2 and MSH3) detect and bind with base-base mismatches and insertion, deletion loops (IDLs), and interact with the N-terminal domain of any MutL homologues such as MutLα (MLH1-PMS2), MutLβ (MLH1-MLH3), and MutLγ (MLH1-PMS1) [113]. The C-terminal of MutL homologues has latent endonuclease activity that recruits proliferating cell nuclear antigen (PCNA), replication factor C subunit 1 (RFC), and exonuclease (EXO1) to perform the excision step. DNA polymerase-δ facilitates high-fidelity DNA synthesis, while DNA ligase I/IV seals the nick [114]. MMR dysfunction results an autosomal-dominant inherited cancer predisposition syndrome (also called Lynch syndrome) that increases the prevalence of sporadic cancers [115]. MSH2 and MSH6 deficient colon cancer cells exhibit increased rates of NO induced mutations compared to MSH2 and MSH6 proficient cell lines, indicating that MMR can play a role in repairing NO related DNA damage [116].

Double stranded breaks (DSBs) are highly deleterious and the most lethal of all DNA lesions. Thus, it is important to understand both HDR- and NHEJ-mediated DSB repairs (Figure 2d). HDR is believed to result in error free DSB repair and healthy cell growth, while NHEJ is error prone and has higher potential for the introduction of malignancy-related mutations. These two pathways are influenced by many regulatory mechanisms [117].

NHEJ is responsible for fixing many two-ended DSBs in eukaryotic cells [118]. NHEJ is initiated with the fast and strong-affinity binding of the Ku70-Ku80 heterodimer (Ku) to DNA ends which prevents DNA end resection and recruits the DNA-dependent protein kinase (DNA-PK) holoenzyme [119]. Subsequently, DNA-PK facilitates the following key functions in NHEJ: (i) activating the DNA nucleases (e.g., ARTEMIS) to process the broken ends and find cohesive nucleotides; (ii) filling the small single-strand gaps in DSB ends by polymerases (e.g., Polμ and Polλ); and (iii) catalysing the DSB ligation by DNA ligase complex consisting of DNA ligase IV, XRCC4, XLF, and PAXX [120,121]. Mutations in NHEJ components cause several abnormal conditions such as severe combined immunodeficiency (SCID), microcephalic primordial dwarfism, and isolated radiation hypersensitivity/malignancy predisposition [122]. Xu et al. reported that NO increases the expression of DNA-PK catalytic subunit thus contributing to DNA-PK activity and NHEJ repair [123].

HDR involves a high-fidelity DSB repair type that plays a major role in DNA repair, DNA replication, meiotic chromosome separation, and telomere conservation [124]. HDR is facilitated by the MRE11-RAD50-NBS1 (MRN) complex, CtIP, and BRCA1. The Mre11-Rad50 (MR) subcomplex shows a dual endonuclease (DNA2) and EXO1 activity to form a short 3′ ssDNA overhang [125]. ssDNA tail degradation by the MRN complex is stopped by binding of replication protein A (RPA) at the 3′ ssDNA tails that create hairpin-capped ends to hinder HR repair [126]. Furthermore, BRCA2 in association with BRCA1 and PALB2 brings RAD51 monomers to ssDNA to remove RPA, resulting in the formation of RAD51 presynaptic filaments to seek out homologous sequences and strand invasion [127]. The invading strand is extended at 3′ end, either by replicative DNA polymerases (POL δ and ε) or translesion DNA polymerases (POL η and κ), resulting in displacement loop (D-loop) formation. D-loop structures can be unravelled by the following homologous repair (HR) pathways: double-strand break repair (DSBR), synthesis-dependent strand annealing (SDSA) or break induced replication (BIR) [128]. NO can impact the activation of the HDR by inhibiting BRCA1 expression via NO activation of PP2A. This leads to RBL2 dephosphorylation, altering RBL2/E2F complexes to favour RBL2/E2F4 complexes which repress the BRCA1 promoter [129]. Intriguingly, Mujoo et al. reported that pluripotent stem cell differentiation is mediated in part by the NO-cGMP pathway. This increases DNA damage, but also represses RAD51 and BRCA1, instead favouring NHEJ with 53BP1 activation [130]. Similar effects of NO have been seen in A549 lung carcinoma cells adapted to chronic NO exposure [131].

### 3.3. Cell Cycle Arrest

NO induced genotoxicity triggers signal transduction cascades that regulate cell cycle checkpoints that allow for activation of restorative DDR machinery. The term checkpoint is defined by the switch between the G1/S, intra-S, and G2/M cell cycle phases [132]. The DNA damage checkpoint is controlled by complex signalling pathways, consisting of three main components: sensors of damage, signal transducers, and effectors that triggers cell cycle arrest, apoptosis, DNA repair, and/or activation of damage induced transcription machinery.

The DNA damage checkpoints are initiated and sensed by essential factors such as Rad9, Rad1, Hus1, and Rad17. The Rad9, Hus1, and Rad1 orthologues structurally resemble proliferating nuclear antigen (PCNA) and create a homotrimeric Rad9–Hus1–Rad1 sliding clamp (9-1-1 complex) that is loaded around the DNA at ongoing DNA replication sites [133]. The RFC complex works as a clamp loader for PCNA, which comprises four small subunits (p36, p37, p38, and p40) and one large subunit (pl40). However, for the 9-1-1 complex, Rad17 interacts with the four small subunits of RFC to make an alternative clamp loader complex (Rad17–RFC) [134]. The chromatin-bound 9-1-1 complex phosphorylates and activates the checkpoint-signalling cascade mediated by ataxia telangiectasia mutated (ATM) and ATM and Rad3 related (ATR) proteins. ATM and ATR are members of the phosphatidylinositol 3-kinase (PI3K)-related kinase (PIKK) family which belong to the serine/threonine protein kinase (270–450 kDa) superfamily that transmit the damage signal to the effector checkpoint kinases 1 and/or 2 (Chk1/Chk2) [135]. In addition, mediator proteins including BRCA1, Claspin, p53 binding protein1 (53BP1), topoisomerase binding protein1 (TopBP1), and mediator of DNA damage checkpoint1 (MDC1) have also been implicated in the DNA damage response [136]. Finally, the phosphorylation of the cell division cycle proteins (Cdc25A, Cdc25B, and Cdc25C) by Chk1/Chk2 or dephosphorylation of Cdc25A–C regulates cyclin/CDK complexes that co-ordinate G1/S, intra-S, and G2/M transitions (Figure 3) [137,138].

Cells exposed to genotoxic agents in early to mid G1 phase may trigger ATM to phosphorylate and activate Chk2 [139]. Activated Chk2 facilitates the phosphorylation of Cdc25A, which causes its ubiquitination and proteasomal degradation [140]. Cdc25A degradation blocks Cdk2 activation preventing Cdc45 loading onto chromatin. The failure of Cdc45 loading onto chromatin stops the recruitment of DNA polymerase α, resulting in cell cycle arrest at the G1/S boundary [141]. Further post-transcriptional modification of p53 is required to maintain this arrest. The nuclear export of p53 and its degradation is inhibited by its phosphorylation on Ser15 by ATM or ATR and on Ser20 by Chk1. Phosphorylation and ubiquitination of ubiquitin ligase Mdm2 also leads to the stabilisation and accumulation of p53 protein in the nucleus [142]. Nuclear accumulation of p53 triggers p21 to bind and suppress the activity of Cdk2/cyclin A/E complexes, thereby inhibiting G1/S transition. In addition, the binding of p21 to the cyclin D–Cdk4 complex suppress the Rb/E2F pathway by phosphorylating Rb protein [143]. When cells experience genotoxicity in S phase, the intra S phase checkpoint interrupts the cell cycle through two different pathways: ATM/ATR–Chk1–Cdc25A and ATM–NBS1–SMC1 [144]. In the first pathway, ATR phosphorylates Chk1 which in turn phosphorylates Cdc25A, resulting its proteasomal degradation. Loss of Cdc25A inactivates cyclin E–Cdk2 and prevents the loading of Cdc45 onto chromatin, thus delaying DNA replication to allow repair [145]. In the second pathway, ATM together with Chk2 activate the Nbs1–Mre11–Rad50 complex and intra-S checkpoint [146]. Other proteins such as 53BP1, BRCA1, and MDC1 also mediate in the intra-S checkpoint activation [147]. NO has context dependent effects on cell cycle control. Under chronic inflammation conditions such as ulcerative colitis (a risk factor for colorectal cancer), NO activation of p53 has been shown to be dependent on ATM and ATR and engages a p53/p21 dependent G2/M checkpoint, which would allow DNA repair of free radical-induced DNA damage [148]. A similar mechanism is seen in neuroblastoma where NO induced an ATR-dependent activation of p53 on Ser15. Of therapeutic relevance was that in this model, neuroblastoma cells were sensitised to irradiation [149].

When DNA damage occurs in G2 phase, the G2/M checkpoint interrupts the cell cycle to stop the cell from entering mitosis (M-Phase). In response to DNA damage, activated ATM transmits two simultaneous cascades that ultimately facilitates the inactivation of the cdc2/Cyclin B complex. In the first cascade, activated Chk1 phosphorylates Cdc25A, which leads to its proteasomal degradation, resulting in the inhibition of Cdc2/cyclin B [150]. In the second cascade, phosphorylation of p53 by activated Chk1 and its dissociation from MDM2 induces its nuclear accumulation and stabilisation, leading to the stimulation of many downstream target genes to block the entry into mitosis including. p53 target genes facilitating this block include 14-3-3 sigma which ties with phosphorylated Cdc2/cyclin B complex to facilitate its nuclear export, GADD45 which induces the dissociation of the Cyclin B-cdc2 complex, and p21 which inhibits a subset of cyclin-dependent kinases such as cdc2 [151]. In addition, different isoforms of p38-MAPK (mainly α and γ) induce the G2/M checkpoint by inhibiting Cdc25B [152]. NO-donors given at therapeutic levels induce cell cycle arrest and apoptosis. The NO-donor JS-K induced G2 cell cycle arrest in HBV-positive hepatocellular carcinoma cells via activation/phosphorylation of ATM/ATR/Chk1/Chk2 and led to caspase activation and apoptosis [153]. Similar effects of JS-K were observed in multiple myeloma [154]. NO-donating aspirin also induces G2/M phase arrest resulting in increased cyclin B1 expression and CDK1 phosphorylation concomitant with decreased cyclin D1 and cdc25 [155].

### 3.4. Apoptotic Effects

Nitrosative DNA damage activates complex signalling networks that induce cells to undergo DNA repair, enter terminal differentiation via senescence, or, if the damage is too severe, undergo apoptosis (Figure 4) [156]. NO can also activate telomerase and delay endothelial cell senescence [157]. Oestradiol and eNOS together regulate telomerase catalytic subunit (hTERT) promoter activity [158]. Cellular apoptosis is regulated by crosstalk between death receptor and mitochondrial signalling pathway. The death receptor or extrinsic pathway regulates cell death facilitated by the binding of specific death ligands to their corresponding cell surface receptors such as Fas, TNFR, CD95, Apo1, and the DR3/4/5, and the receptor for TRAIL. Ligand binding to the trimerised receptor at the cell surface triggers downstream pathways via the recruitment of intracellular adapter molecules like FADD (fas-associated death domain protein) and TRADD (TNF-related death domain protein). These adapter molecules transmit death signals by stimulating proteolytic enzymes such as cysteine proteases (e.g., caspases), which are essential for the cleavage of different intracellular substrates and DNA degradation. In the mitochondrial/intrinsic pathway, secondary messengers such as NO and pro-apoptotic proteins (Bax, Bak, Bid, and caspases) lead to mitochondrial membrane permeabilisation through the permeability transition pore (PTP) complex that allows the subsequent release of mitochondrial cytochrome c and other pro-apoptotic molecules such as SMAC/DIABLO, AIF (apoptosis inducing factor), and intra-mitochondrial caspases into the cytoplasm. Cytochrome c then associates with the adapter protein apoptotic protease activating factor (Apaf1) and induces formation of the Apoptosome complex in the cytoplasm which cleaves procaspase-9 to active caspase-9. Caspase-9 in turn triggers procaspase-3 and other executioner caspases that induce apoptotic death by cleavage of other death substrates [40].

Whether NO induces pro- or anti-apoptotic responses depends on the source of NO, its concentration, and the duration of NO flux [159]. Low levels of NO inhibit apoptosis by activating guanylyl cyclase (GC), inducing the heat shock protein 70 (Hsp 70) response and inhibiting pro-apoptotic proteins such as Bax, in several mammalian cells, including endothelial cells, neural cells, pancreatic islets and many tumour cells (e.g., human ovarian cancer cells) [160,161]. NO exerts most of its anti-apoptotic physiological responses by cGMP-dependent mechanisms or post-translational modifications that upregulate the intracellular antioxidant system (such as glutathione), inactivating caspases, and other apoptotic proteins [162,163]. cGMP can directly regulate cAMP response element (CRE)-binding protein (CREB) through phosphorylation by cGMP-dependent protein kinase G (PKG) or crosstalk with other signalling pathways such as MAPK, calcineurin, and RhoA pathways [164].

In contrast, high levels of NO induce apoptosis by activating apoptosis-related receptors such as Fas (also known CD95 or APO-1), death receptor 5, and TNFR1 (also known as p55 or CD120a) [165]. Fas activation leads to cellular apoptosis [166]. TNFR1 activation is more complex. In response to NO-mediated DNA damage, SUMOylation and ATM-dependent NEMO phosphorylation (NFκB essential modulator) activates the expression of NFκB-mediated FAS ligand and receptor-interacting protein 1 (RIP1). This leads to NK3-mediated IL8 secretion and recruitment of FADD to trigger caspase 8 activity, resulting in apoptosis [167]. In addition, NO-mediated activation of MAPK pathways including p38-kinase, JNK (c-Jun N-terminal kinase), and ERK1/2 (extracellular regulated kinases) can result in p53 accumulation, caspase activation and cell death [168,169,170]. Alternatively, activation of the TNFR can lead to NFκB activation resulting in pro-survival signalling and activation of pro-inflammatory cytokines [171].

### 3.5. Angiogenic Effect

Angiogenesis is tightly controlled by the balance between pro-angiogenic and anti-angiogenic factors [172]. In normal circumstances, the balance favours the anti-angiogenic factors, inhibiting angiogenesis. In contrast, during tumour progression, pro-angiogenic factors such as vascular endothelial growth factor (VEGF) and prostaglandins are highly expressed and anti-angiogenic factors such as thrombospondin-1 (TSP1) or endostatin are inhibited (Figure 5) [173].

Most NO-driven angiogenic signals are facilitated at low NO fluxes derived from eNOS that play a vital role in cardiovascular function and blood pressure regulation in the body and regulate blood vessel formation and remodelling in pregnancy [174,175]. NO/eNOS also regulates tumour angiogenesis and wound healing by inducing several angiogenic factors. The reduced supply of oxygen (hypoxia) and nutrients in rapidly proliferating cells of solid tumours stimulates neovascularisation by promoting the expression of hypoxia-inducible factor-1α (HIF-1α). HIF-1α is stabilised by posttranslational modification of the protein in hypoxic conditions and which allows it to complex with its constitutively expressed β subunit. This heterodimer then translocates to the nucleus where it recruits coactivators CBP/p300 to various promoters containing hypoxia response elements (HREs), leading to the activation of many target genes including *VEGF* [176]. VEGF is one of the most potent pro-angiogenic factors which mediates the phosphorylation of eNOS at the serine 1177 residue by AKT and a consequent increase in NO production [177,178,179].

NO further promotes HIF-1α stabilisation enhancing the effects of hypoxia in promoting tumour neoangiogenesis. Hypoxia can also facilitate the recruitment of Akt to Ca^2+^/CaM-activated eNOS by the binding of hsp90, thereby leading to its enzymatic phosphorylation and a long-lasting NO release, even in conditions of low Ca^2+^ [180,181]. This sustained NO-flux can also inhibit anti-angiogenic thrombospondin-1 (TSP1) and activate the NO/cyclic guanosine monophosphate (cGMP) pathway via ERK phosphorylation within the endothelial compartment to stimulate neovascular growth [182,183]. Prostaglandin E2 (PGE2) stimulates angiogenesis by enhancing eNOS phosphorylation and triggering the NO/cGMP pathway in endothelial cells of the human umbilical vein [184]. The endothelium-mediated relaxation and the smooth muscle contraction of the rabbit pulmonary artery in response to [Sar9]-SP-sulphone (a stable and selective agonist for the tachykinin NK1 receptor) and prostaglandin E1 (PGE1) was enhanced by sodium nitroprusside (SNP) and abrogated by the eNOS inhibitor, L-NAME [185].

### 3.6. Epithelial-to-Mesenchymal Transition (EMT) and Metastatic Effects

Epithelial-to-mesenchymal transition (EMT) is essential to transform benign tumours into aggressive and highly invasive cancers in which the stationary epithelial cells develop the ability to migrate and invade through the surrounding tissue into blood or lymphatic vessels (intravasation), exit the vessel and relocate at distant sites (extravasation), and generate secondary tumour masses at new sites (Figure 5) [186,187]. The EMT process is governed by the downregulation of epithelial markers, e.g. E-cadherin, and upregulation of mesenchymal markers, e.g., vimentin [188,189].

A high flux of NO prevents NF-κB activity either by inhibiting the phosphorylation and subsequent loss of IκBα (which prevents the nuclear localisation of NF-κβ), or S-nitrosation of the p50 subunit of NF-κB to reduce its DNA binding activity [190]. Reduced activity of NF-κB also modulates EMT outputs, by controlling downstream EMT-related markers such as Snail, Yin Yang (YY) 1, Raf kinase inhibitor protein (RKIP), phosphatase and tensin homologue (PTEN). High levels of NO can directly inhibit Snail, a master regulator of the EMT transcriptional programme). This leads to the derepression of its targets, e.g., RKIP and E-cadherin, resulting in the suppression of EMT and metastasis [191]. NO-mediated metastasis suppression was demonstrated in vitro by its observed enhancement of the expression of N-myc downstream-regulated gene 1 (NDRG1) in HCC1806 breast cancer cells. This anti-metastatic effect was regulated by incorporation of chelatable iron into a dinitrosyliron complex [192]. Resistant tumour cells are also sensitised by NO to modulate the apoptotic stimuli from chemotherapeutic agents [191].

NO has been shown to attenuate several transcription factors (SNAI1, SNAI2, ZEB1, and ZEB2), growth factors (EGF, PDGF, HGF) and other signalling molecules like TGF-β, sonic hedgehog (Shh), Wnt/beta-catenin and extracellular matrix (ECM) components which maintain the epithelial morphology and suppress cell migration and invasion [193,194,195]. NO has also been reported to suppress tumour growth and metastases in different in vivo and in vitro cancer models. Topical application of NO-exisulind in ultraviolet B-induced skin cancer murine model suppressed EMT by inducing E-Cadherin and inhibiting N-cadherin, fibronectin, SNAIL, Slug and Twist [196]. NO can also attenuate matrix metalloproteinase (MMP, a family of “pro-metastatic” enzymes) activity that is responsible for the degradation of the basement membrane necessary for tissue invasion and tumour cell dissemination [197]. NO can suppress the mRNA-stabilising factor HuR, resulting in the destabilisation of MMP-9 mRNA [198]. IL-2/α-CD40–induced NO repress the activity of MMP9 and concomitantly increased the expression of E-cadherin and tissue inhibitor of metalloproteinase (TIMP) 1 in an orthotopic model of renal cell carcinoma [199]. NO attenuated 12-*O*-tetradecanoylphorbol 13-acetate (TPA) induction of MMP-9 by inhibiting protein Kinase C (PKC) in breast cancer [194]. Transfection of iNOS induces apoptosis and suppresses tumour progression and metastasis in both ectopic and orthotopic xenograft mouse models of pancreatic cancer [200]. The anti-cancer activity of *Rhus coriaria* has been demonstrated by its attenuation of NFκB, STAT3, and NO pathways that suppress cell migration and invasion and induce the cell cycle arrest and cell death in triple negative breast cancer (MDA-MB-231) cells [201]. Tumour growth and metastasis was decreased in iNOS transfected B88 oral squamous carcinoma cells in both in vivo and in vitro settings [202]. In contrast, the iNOS null fibrosarcoma cell line (KX-dw1, KX-dw4, and KX-dw7) and M5076 murine ovarian sarcoma cell lines injected into iNOS(_¯_/_¯_) mice showed more tumourigenicity and lung metastasis as compared to iNOS(^+^/^+^) counterparts [203,204]. Treatment of TGF-β1-induced alveolar cells with DETA/NO (NO-donor) reduced EMT while treatment with L-NAME (NOS inhibitor) caused a spontaneous increase in EMT [193].

Promigratory effects of NO have also been reported. NO inhibits the aggregation of tumour cells and platelets in the microcirculation via cGMP-dependent mechanisms [205]. NO activates EGFR and Src via *S*-nitrosation, leading to activation of β-catenin signalling in ER-negative and basal-like breast cancer. Furthermore, treatment with an NO-donor (DETA/NO) of ER-negative breast cancer cells induced EMT by decreasing expression of E-cadherin and a concomitant increase in vimentin and β-catenin [206]. Spontaneously developing tumours in a murine mammary adenocarcinoma (C3H/HeJ) model showed heterogeneous expression of eNOS within the primary tumours and homogeneous eNOS positivity in their metastatic counterparts, suggesting NO mediated tumour progression and metastasis. Furthermore, higher expression of eNOS was observed in clones derived from a spontaneous mammary tumour C3L5 (highly metastatic) than C10 cells (weakly metastatic). However, treatment of C3L5 and C10 bearing tumour mice with an eNOS inhibitor (L-NAME) reduced invasiveness of both cell lines [207,208]. B16-BL6 murine melanoma cells injected in syngeneic iNOS(^+^/^+^) mice produced more and larger metastases than their iNOS(_¯_/_¯_) counterparts [204]. A threefold higher invasive potential was observed in HRT-18 cells constitutively expressing iNOS compared to non-iNOS expressing HT-29 cells using Matrigel invasion assay. Treatment of HT-29 cells with an NO-donor (DETA/NO, 50 nM) and inflammatory cytokines (IFN-γ and IL-1α) in an independent experiment showed a significant increase in invasiveness, whereas the invasiveness of HRT-18 and HT29 were partially inhibited by the NOS inhibitor 1400W [209]. NO produced by iNOS enhanced the activity of MMP-9, resulting in angiogenesis, invasion, and metastasis in hepatocellular carcinoma [210]. MMP-9 and/or uPAR activated by iNOS in glioma cells induced cell migration via their interaction with α9β1 integrin [211].

### 3.7. Immunomodulatory Effects

NO-mediated communication between tumours and the microenvironment can have a major impact on tumour biology. NO regulates apoptosis and survival of different immune cells (dendritic cells, mast cells, NK cells, and phagocytes such as macrophages, monocytes, Kupffer cells, microglia, eosinophils, and neutrophils) and other cells in the tumour microenvironment (epithelial cells, endothelial cells, vascular smooth muscle cells, Schwann cells, fibroblasts, keratinocytes, chondrocytes, hepatocytes, and mesangial cells) [212,213]. Induction of the innate immune response is initiated by activation of classical macrophages (M1) that secrete proinflammatory cytokines (e.g., TNF-α, IL-1β, and IL-6), proteases (e.g., MMP-9), and NO/RNS. Increased production of NO activates downstream signalling pathways that perform a critical role in the cytotoxic activity of immune cells against tumour cells [214]. Similarly, the tumouricidal and pathogen eradication capacity of natural killer (NK) cells is regulated partly by their NO synthesising ability [215].

Even though macrophage-generated NO participates in cytotoxic and antitumour activity in the tumour microenvironment, NO is recognised as an immunosuppressive mediator that helps to create a barrier against the anti-tumour immune responses [216]. NO-mediated immunosuppression has been demonstrated in human prostate carcinoma by its inhibition of arginase 1 (ARG1) in T cells and diminished antitumoural activity of tumour-infiltrating lymphocytes (TILs) resulting from tyrosine nitration [217]. NO production may also lead to *S*-nitrosation of the chemokine CCL2 and inhibit the infiltration of T cells into the tumour through its ability to retain myeloid-derived suppressor cells (MDSCs) [218]. NO produced by MDSCs reduces endothelial expression of E-selectin in human squamous cell carcinomas, thus preventing the recruitment T-cells into tumours [219,220]. The autocrine effects of iNOS in IL-17 secreting γδT cells plays a key role in the recruitment of MDSCs and activation of regulatory T (Treg) cells that induce the pro-tumourigenic physiognomy in melanoma [221,222]. MDSC-derived NO inhibits antigen presentation to CD4+ helper T cells by dendritic cells (DCs) and nitrates STAT1 to impair of T-cell JAK/STAT signalling proteins necessary for various T-cell functions in melanoma tumour models [223]. MDSC-derived NO has also been demonstrated to lead to nitration of the tyrosine residues of NK cell-specific proteins, impairing Fc receptor-mediated NK cell functions and therapeutic responses to monoclonal antibodies [224]. Thus, NO exerts widely different impacts on the recruitment of myeloid and lymphoid-derived cells, thereby restricting the capacity of iNOS inhibition to restore immune function [225]. In considering the role of NO in immunomodulation it is also important to note that species specific differences exist in NO fluxes from iNOS, which may impact the roles of iNOS in immune cell modulation by different cell types including mesenchymal stromal cells and macrophages [226,227,228]. A more detailed review on the role of NO in macrophages and cancer is provided in [229].

## 4. NO-Mediated Strategies for Cancer Treatment

Due to the biphasic nature of NO signalling, modulation of NO biosynthesis and exogenous delivery have been purported as therapeutic strategies against cancer [230]. While tumour cell derived NO is rarely capable of killing cancer cells, it can be exploited to achieve clinical benefit by sensitising tumour cells to chemo, immune- and radiotoxicities [231]. Thus, releasing adequate amounts of NO at the cancer site by iNOS over-expression (gene therapy) or administration of NO-donors (such as organic nitrates, *N*-nitrosamines, *S*-nitrozotiols, nitrozimines, and metal–NO complexes) or administration of NO-donor linked non-steroidal anti-inflammatory drugs (NO-NSAIDs) represent attractive novel strategies for cancer treatment [232].

Many NO-donors have been generated that can be defined on the basis of their utilisation in cancer therapeutics. Organic nitrates such as glyceryltrinitrate (GTN) and isosorbidedinitrate (ISDN) are the oldest class of NO-donors. The anticancer properties of GTN have been demonstrated in both in vitro and in vivo murine melanoma models. GTN has also being used with vinorelbine/cisplatin as a chemosensitiser in human NSCLC [233]. Due to the high affinity of NO for metals, metal-NO complexes including sodium nitroprusside (SNP) have been synthesised as anti-neoplastic compounds with activity in prostate, gastric and cervical cancer cells, and radiosensitisating ability in pancreatic and glioma cancer cells [234]. *S*-nitrosothiols (RSNO) such as S-nitroso-*N*-acetylpenicillamine (SNAP) and S-nitrosoglutathione (GSNO) are also used as anti-cancerous agents. Both SNAP and GSNO are relatively stable and have anti-neoplastic impacts in different malignancies. In fact, both compounds have been shown to act significantly in the radiosensitisation of different tumours [235,236]. Sydnonimines are another class of NO-releasing mesoionic heterocyclic compounds that yield superoxide-generated peroxynitrite during their decomposition [237]. 3-morpholinosydnonimine (SIN-1) has been the most widely investigated compounds of this class. SIN-1 causes single-stranded DNA breaks, stimulates protein nitration and suppresses mitochondrial respiration [60,238]. Diazeniumdiolates (also called NONOates) are a class of NO-donors that have gained tremendous attention due to release of controlled fluxes of NO in both in vitro and in vivo applications. Such compounds have a common structure, R^1^R^2^NN(O)=NOR^3^, in which all three R groups can be varied to create a large series of interesting compounds [239].

Depending on the identity of the amine group, such compounds have half-lives ranging from 2 s to 20 h. These compounds have been widely tested as chemo- and radiosensitising drugs/agents to produce synergistic results with traditional cancer therapeutics [234]. Hybrid NO-donor drugs have been developed by hybridising the NO-donor(s) with an anticancer drug/agent/fragment, ideally by a cancer-specific linker of synergistic function that reduces the adverse effects caused by either component alone. This rationale triggered the development of various NO-linked NSAIDs (NO-NSAIDs) in which the NO-releasing moiety was linked by covalent bonding to standard NSAIDs such as salicylic acid, aspirin, indomethacin, sulindac or ibuprofen. The NSAID-linker-NO-donor adduct is often separated by non-specific esterase activity, which results in synergistic COX-inhibition and NO-production [240]. Various NO-NSAIDs are identified as NCX-# (i.e., NCX-976, NCX-4016, NCX-4040, and NCX-4215) and are currently being explored in cancer therapy.

In the case of tumours that overexpress iNOS leading to increased metastasis and poor patient outcomes, an approach utilising iNOS inhibitors may be appropriate. This is particularly relevant to cancers where iNOS promotes poor outcomes. iNOS is known to be associated with poor outcome in ER negative breast cancer [241], and the rarer triple negative breast cancer (TNBC) [242]. In a nude mouse model of human triple negative breast cancer with the cell line MDA-MB-231, the iNOS inhibitor aminoguanidine inhibited tumour growth and metastasis to the brain. This was accompanied by the repression of tumour promoting factors including S100A8, IL-6, and IFN-γ [243]. Dual inhibition of iNOS (aminoguanidine) and COX2 (aspirin), which predict poor outcome in ER negative breast cancer, further repressed MDA-MB-231 tumour growth via repression of TRAF2 dependent signalling [244]. Metaplastic breast cancer is a stromal dominant rare form of breast cancer, with a high association with the triple negative breast cancer subtype. Elevated iNOS and ribosomal protein L39 was associated with poor survival in metaplastic tumours, while the pan-NOS inhibitor N^G^-methyl-L-arginine acetate (L-NMMA) repressed metaplastic murine model tumour growth [245]. Further exploration of iNOS in triple negative breast cancer patients revealed that iNOS activates EGFR and MEK-ERK signalling, leading to increased risk of distant metastasis and poor outcome [246]. Dávila-González et al. investigated the ability of t L-NMMA to repress TNBC patient-derived xenografts tumour growth, and found that combination therapy with docetaxel ameliorated tumour growth via activation of ATF4-CHOP mediated apoptosis [247]. Similarly iNOS inhibition reduced tumour growth in a murine model of Kras and p53 mutation-positive non-small cell lung cancer, and enhanced the efficacy of carboplatin based chemotherapy [248]. The NOS inhibitor N-nitro-l-arginine repressed tumour growth in a murine pancreatic cancer model. When given in combination with a VEGFR2 inhibitor it further repressed tumour growth and vascular perfusion [249]. Similar effects have been observed in pancreatic cancer with eNOS specific inhibitors [250] and pan-NOS inhibitors [251].

## 5. Conclusions

The evidence summarised in this review demonstrates that NO plays a crucial role in the regulation of tumour growth, angiogenesis, and metastasis. The role of NO in tumour biology depends on its source and thus its spatial, temporal, and multi-level dose control. Taken together, this determines a pro- or anti-tumourigenic response mediated by a cGMP-dependent process and/or post-translational protein alterations. Given that NO has the ability to initiate or regulate virtually all the hallmarks of cancer, this suggests that NO may be a master regulator of tumourigenesis and tumour promotion. This presents an opportunity to disrupt aberrant NO signalling at various points along the carcinogenesis pathway and thus simultaneously target multiple hallmarks of cancer. Thus, the findings in both in vitro and in vivo experiments related to NO have now made a strong argument that NO alone or in combination with anticancer drugs may provide new routes for the treatment of a variety of human malignancies. Although these findings have yet to be fully translated to the clinic, clinical trials of NO-donors and iNOS inhibitors are currently underway. The application of NO-based anticancer drugs certainly merits further investigation to harness and unlock its potential as a novel therapeutic strategy.

## Figures and Tables

**Figure 1 ijms-21-09393-f001:**
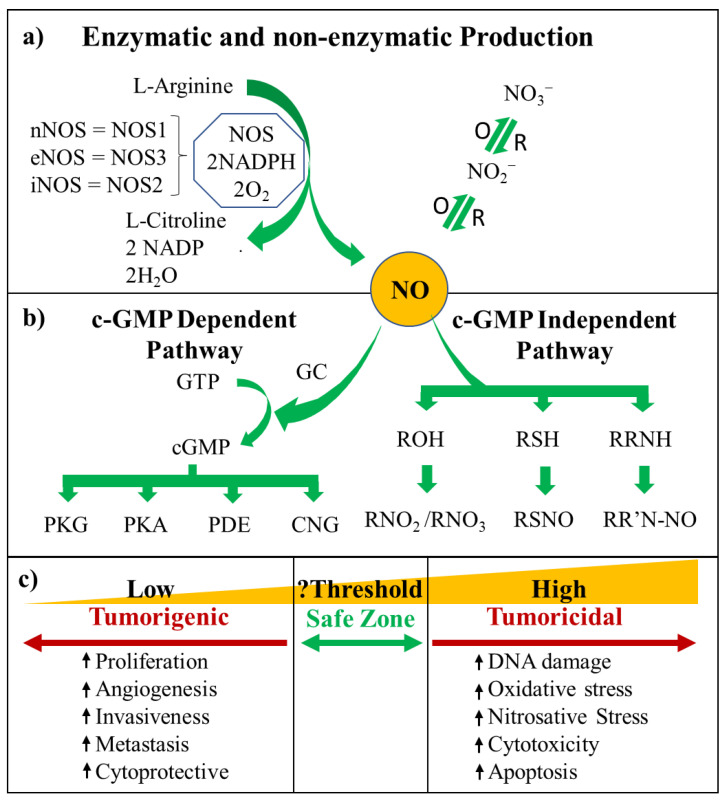
NO formation, its mechanism of action, and its phenotypic consequences: (**a**) Enzymatic and non-enzymatic synthesis of NO and facilitation of nitrogen oxide interchange. (**b**) Downstream signalling events induced by NO. The cGMP-dependant pathway shows the interaction of NO with soluble guanylate cyclase (sGC) which catalyses the conversion of GTP to cGMP. cGMP then triggers protein kinase G (PKG), protein kinase A (PKA), phosphodiesterases (PDE), and ion gated channels (CNG). The cGMP-independent pathway involves the posttranslational modification of proteins containing, thiols (RSH), and amines (RR’NH), which react with RNS formed from NO to produce nitrite/nitrate (RO-NO), S-nitrosothiols (RSNO), and N-nitrosoamines (RR’NNO), respectively. (**c**) Concentration-dependent effects of NO in cancer. Low levels produce pro-tumourigenic effects while high NO is predominantly anti-tumour. Created with BioRender.com.

**Figure 2 ijms-21-09393-f002:**
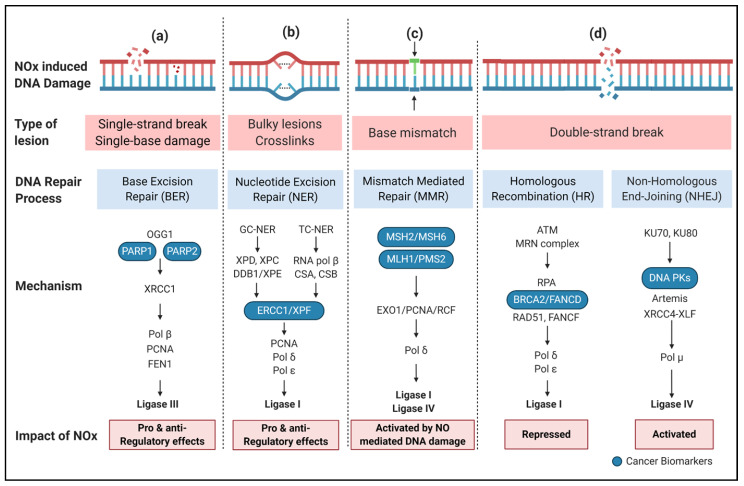
Effect RNS- induced DNA damage on DNA DAMAGE Repair (DDR) pathways: (**a**) RNS can have both pro- and anti-regulatory effects on base excision repair (BER) mechanisms in response to single strand DNA breaks; (**b**) RNS can have both pro- and anti-regulatory effects on nucleotide excision repair (NER) mechanisms in response to bulky DNA adducts; (**c**) mismatch mediated repair (MMR) mechanisms are activated in response to NO related DNA damage; and (**d**) non-homologous end joining repair (NHEJ) is favoured over homologous recombination (HR) in response to NO induced double stranded DNA breaks. Created with BioRender.com.

**Figure 3 ijms-21-09393-f003:**
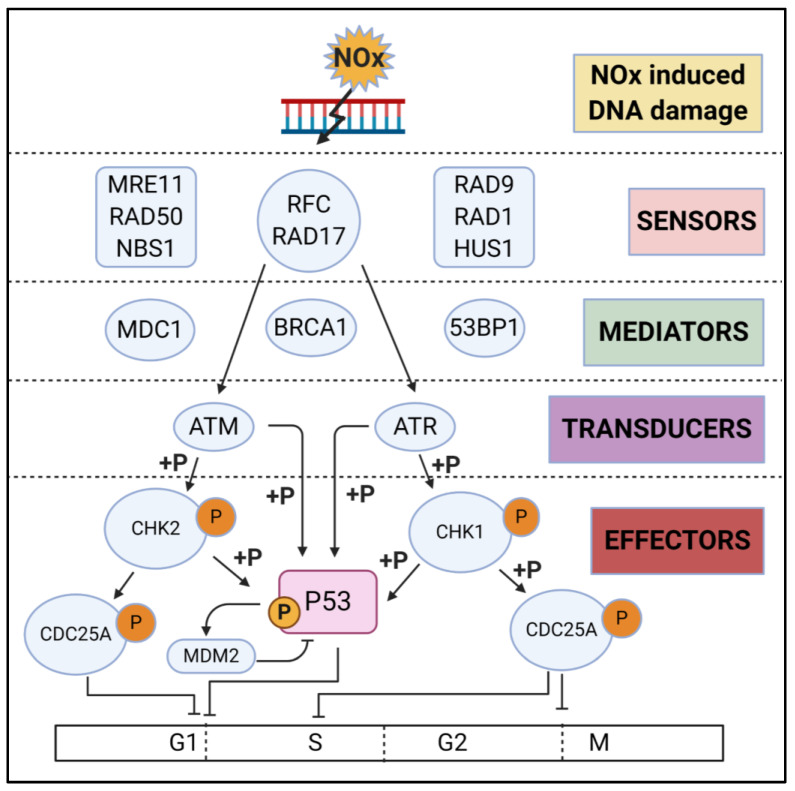
Cell cycle check point responses. RNS-induced DNA damage is sensed by a multitude of DNA damage sensors which relay the signal to the DNA damage transducers ATM and ATR. RNS-induced DNA damage can lead to arrest during different phases of the cell cycle depending on the type of DNA damage and the effector signal activated. Created with BioRender.com.

**Figure 4 ijms-21-09393-f004:**
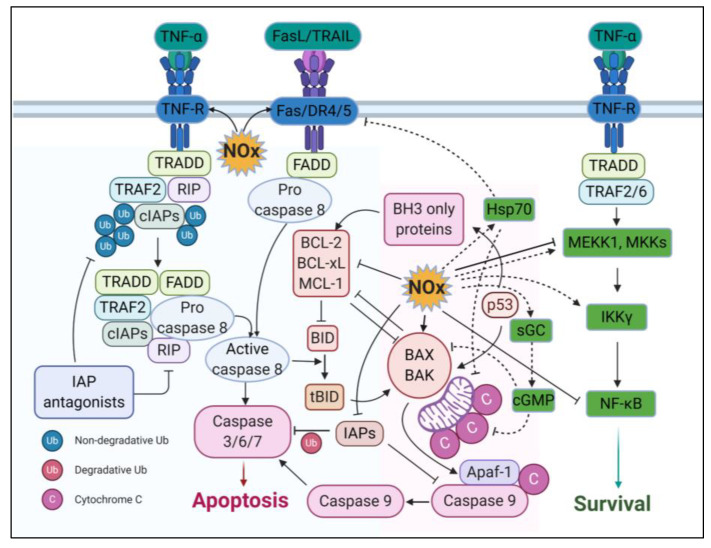
Apoptosis induced by high levels of NO is mediated via activation of the extrinsic apoptosis pathway which involves the TNFR and Fas/Death Receptors. In contrast, low levels of NO repress the intrinsic apoptosis pathway in a cGMP dependent manner (low NO, dashed line; high NO, solid line). Created with BioRender.com.

**Figure 5 ijms-21-09393-f005:**
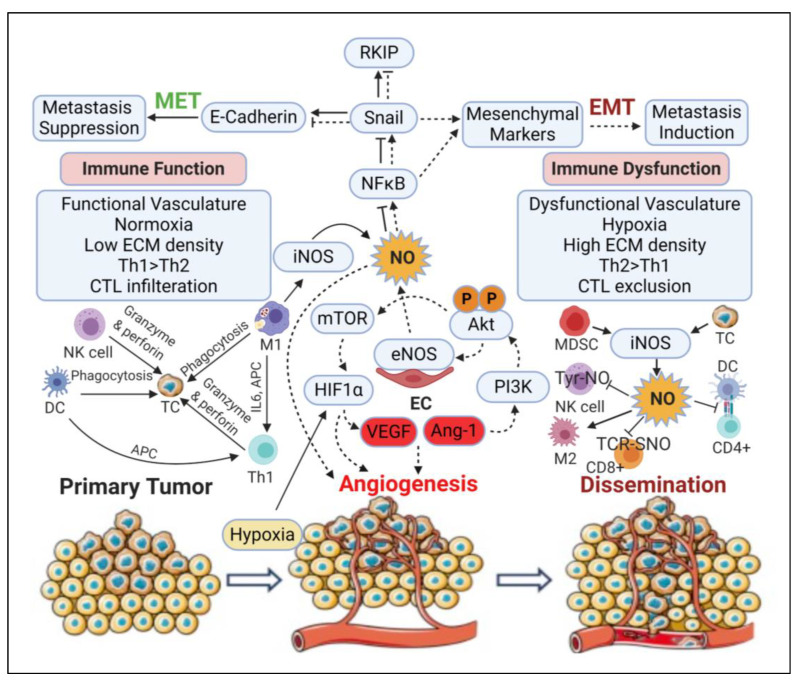
Low to moderate levels of NO induce SNAIL and lead to epithelial to mesenchymal transition, increased angiogenesis and immune dysfunction promoting tumour progression and dissemination. High levels of NO (e.g., from M1 macrophages or Th1 lymphocytes) lead to tumour toxicity, repress EMT and mount an anti-tumour immune response. (low NO, dashed line; high NO, solid line) Created with BioRender.com.

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
