# Peer review of "The Role of Nitric Oxide in Cancer: Master Regulator or NOt?"

_ijms, 2020, doi:10.3390/ijms21249393_

Round 1

Reviewer 1 Report

This review is a well written review; it is thorough, updated, and covers all the known controversial aspects of NO. 

I have minor concerns: 

  1. Abstract sentence:  " However, there is a 'fine line'  with high levels of NO  (derived from inducible nitric oxide synthase (iNOS) expressing M1 and Th1 polarised macrophages and lymphocytes but .......................with the caveat that the existing evidence is mainly based on murine iNOS  which produces higher fluxes of NO than human iNOS"               This abstract sentence  is not clear or perhaps too long.   Additionally, it was not clear where the murine iNOS versus human iNOS has been made in the text in terms of flux, although  known. This long sentence may be divided for clarity. 
  2. The conclusion or a paragraph before conclusion  may benefit with a clearer section of authors' succinct opinion or a proposal of master regulatory role.  
  3. 3. Immunomodulation via NO:  Is M1 and M2 transition/ or  interconversion a possibility via  NO? 

Author Response

Response to Reviewer 1

REVIEWER Point 1: Abstract sentence:  " However, there is a 'fine line'  with high levels of NO  (derived from inducible nitric oxide synthase (iNOS) expressing M1 and Th1 polarised macrophages and lymphocytes but .......................with the caveat that the existing evidence is mainly based on murine iNOS  which produces higher fluxes of NO than human iNOS"               This abstract sentence  is not clear or perhaps too long.   Additionally, it was not clear where the murine iNOS versus human iNOS has been made in the text in terms of flux, although  known. This long sentence may be divided for clarity. 

Response to Point 1:  We thank the reviewer for pointing out the confusion in this sentence in the abstract. We have replaced the text in the abstract with the following (Lines 32-35):

“In contrast high levels of NO derived from inducible nitric oxide synthase (iNOS) expressing M1 and Th1 polarised macrophages and lymphocytes may exert an anti-tumour effect protecting against cancer. It is important to note that the existing evidence on immunomodulation is mainly based on murine iNOS studies which produce higher fluxes of NO than human iNOS.”

We have also added this additional text at Line 601-605 to clarify this point:

“In considering the role of NO in immunomodulation it is also important to note that species specific differences exist in NO fluxes from iNOS, which may impact the roles of iNOS in immune cell modulation by different cell types including mesenchymal stromal cells and macrophages {Su, 2014 #4136}{Chinnadurai, 2019 #4137}{Hoos, 2014 #4138}. A more detailed review on the role of NO in macrophages and cancer is provided in {Ryan, 2015 #3458}.”

REVIEWER Point 2:  The conclusion or a paragraph before conclusion may benefit with a clearer section of authors' succinct opinion or a proposal of master regulatory role.  

Response to Point 2:   We have added the following text to the conclusion on line “676-679”:

Given that NO has the ability to initiate or regulate virtually all the hallmarks of cancer, this suggests that NO may be a master regulator of tumourigenesis and tumour promotion. This presents an opportunity to disrupt aberrant NO signalling at various points along the carcinogenesis pathway and thus simultaneously target multiple hallmarks of cancer.

REVIEWER Point 3:. Immunomodulation via NO:  Is M1 and M2 transition/ or  interconversion a possibility via  NO?   

Response to Point 3: We have preliminary data suggesting that it is, however this is not yet published.  In looking at the literature this appears to be as of yet unknown.  We have however added above in our response to Point 3 - A more detailed review on the role of NO in macrophages and cancer is provided in {Ryan, 2015 #3458}. – which gives a good overview of the role of NO in macrophages with respect to cancer.

Reviewer 2 Report

This is the large review  presenting the involvement of NO in cancer biology and its putative treatment. The authors present the various effects  of NO through  cGMP-dependent  and cGMP-independent mechanisms  on the processes of tumorigenesis, like DNA damage, cell cycle,apoptosis, angiognesie and metastasis. Finally, they also  present the preclinical attempts of NO-based  cancer treatments. The presentation of the literature data  concerning the problem contains all the most important contributions. The review can be published in the present form.

Author Response

We thank Reviewer 2 for their kind comments.